# Interpreting Embedding Spaces by Conceptualization

**Adi Simhi, Shaul Markovitch**
The Henry and Marilyn Taub Faculty of Computer Science
Technion – Israel Institute of Technology
`{adi.simhi,shaulm}@cs.technion.ac.il`

## Abstract

One of the main methods for computational interpretation of a text is mapping it into a vector in some embedding space. Such vectors can then be used for a variety of textual processing tasks. Recently, most embedding spaces are a product of training large language models (LLMs). One major drawback of this type of representation is their incomprehensibility to humans. Understanding the embedding space is crucial for several important needs, including the need to debug the embedding method and compare it to alternatives, and the need to detect biases hidden in the model. In this paper, we present a novel method of understanding embeddings by transforming a latent embedding space into a comprehensible conceptual space. We present an algorithm for deriving a conceptual space with dynamic on-demand granularity. We devise a new evaluation method, using either human rater or LLM-based raters, to show that the conceptualized vectors indeed represent the semantics of the original latent ones. We show the use of our method for various tasks, including comparing the semantics of alternative models and tracing the layers of the LLM. The code is available online[1].

## 1 Introduction

Recently, there has been significant progress in Natural Language Processing thanks to the development of Large Language Models (LLMs). These models are based on deep neural networks and are trained on extensive volumes of textual data (Devlin et al., 2019; Raffel et al., 2020; Liu et al., 2019).

While these powerful models show excellent performance on a variety of tasks, they suffer from a significant drawback. Their complex structure hinders our ability to understand their reasoning process. This limitation becomes crucial in several important scenarios, including the need to explain the decisions made by a system that employs the model, the necessity to debug the model and compare it with alternatives, and the requirement to identify any hidden biases within the model (Burkart and Huber, 2021; Ribeiro et al., 2016b; Madsen et al., 2022).

Current LLMs process text by projecting it into an internal embedding space. By understanding this space, we can therefore gain an understanding of the model. Such understanding, however, is challenging as the dimensions of the embedding space are usually not human-understandable.

The importance of interpretability has been recognized by many researchers. Several works present methods for explaining the *decision* of a system that uses the embedding (mainly classifiers) (e.g. Ribeiro et al., 2016a; Lundberg and Lee, 2017a). Some works (Senel et al., 2022; Faruqui et al., 2015) perform training or retraining for generating a *new* model that is interpretable, thus detouring the problem of understanding the original one. Another line of work assumes the availability of an embedding matrix and uses it to find orthogonal transformations, such that the new dimensions will be more understandable (Dufter and Schütze, 2019; Park et al., 2017). Probing methods utilize classification techniques to identify the meaning associated with individual dimensions of the original embedding space (Clark et al., 2019; Dalvi et al., 2019).

In this work, we present a novel methodology for interpretability of LLMs by conceptualizing the original embedding space. Our algorithm enables the mapping of any vector in the original latent space to a vector in a human-understandable conceptual space. Importantly, our approach does not assume that latent dimension corresponds to an explicit and easily-interpretable concept.

Our method can be used in various ways:

---

1. Given an input text and its latent vector, our algorithm allows understanding of the semantics of it *according to the model*.

2. It can help us to gain an understanding of the model, including its strengths and weaknesses, by probing it with texts in subjects that are of interest to us. This understanding can be used for debugging a given model or for comparing alternative models.

3. Given a decision system based on the LLM, our algorithm can help to understand the decision and to explain it using the conceptual representation. This can also be useful in detecting biased decisions.

Our contributions are:

1. We present a model-agnostic method for interpreting embeddings, which works with any model without the need for additional training. Our approach only requires a black box that takes a text fragment as input and produces a vector as output.

2. We present a novel algorithm that, given an ontology, can generate a conceptual embedding space for any desired size and can be selectively refined to specialize in specific subjects.

3. We introduce a new method for evaluating algorithms for embedding interpretation using either a human or an LLM-based evaluator.

## 2    The Conceptualization Algorithm

Let $T$ be a space of textual objects (sentences, for example). Let $L = L_1 \times \ldots \times L_k$ be a latent embedding space of $k$ dimensions. Let $f : T \to L$ be a function that maps a text object to a vector in the latent space. Typically, $f$ will be an LLM or LLM-based.

Our method requires two components: A set of concepts $C = c_1, \ldots, c_n$ defining a conceptual space $\mathcal{C} = c_1 \times \ldots \times c_n$, and a mapping function $\tau : C \to T$ that returns a textual representation for each concept in $C$.

In the pre-processing stage, we map each concept $c \in C$ to a vector in $L$ by applying $f$ on $\tau(c)$, the textual representation of $c$. We thus define $n$ vectors in $L$, $\widehat{c}_1, \ldots, \widehat{c}_n$ such that $\widehat{c}_i \equiv f(\tau(c_i))$.

Given a vector $l \in L$ (that typically represents some input text), we measure its similarity to each vector $\widehat{c}_i$ (that represents the concept $c_i$) using any given similarity measure $sim$. The algorithm then outputs a vector in the conceptual space, using the similarities as the dimensions.

We have thus defined a meta-algorithm CES (Conceptualizing Embedding Spaces) that, for any given embedding method $f$, a set of concepts $C$ and a mapping function $\tau$ from concepts to text, takes a vector in the latent space $L$ and returns a vector in the conceptual space $\mathcal{C}$:

$$\text{CES}^{f,C,\tau}(l) = \langle sim(l, \widehat{c}_1), \ldots, sim(l, \widehat{c}_n) \rangle^T$$

A graphical representation of the process is depicted in Figure 1.

If we use cosine similarity as $sim$, and use a normalised $f$ function, we can implement CES as matrix multiplication, which can accelerate our computation. First, observe that, under these restrictions, cosine similarity is equivalent to the dot product between vectors. Let $U = u_1, \ldots, u_k$ be the standard basis in $k$ dimensions as a base of $L$. We can look at the projection of $U$ in the $\mathcal{C}$ space, by using function $\phi$ such that $\phi(u_i) = \langle \phi(u_i^1), \ldots, \phi(u_i^n) \rangle^T$ where $\phi(u_i^j) = cosine(u_i, c_j) = u_i \cdot \widehat{c}_j$. We can now create a $n \times k$ matrix $M = \langle \phi(u_1), \ldots, \phi(u_k) \rangle$. Using this matrix, we get $\text{CES}^{f,C,\tau}(l) = M \cdot l$.

### 2.1    Generating Conceptual Spaces

To allow a conceptual representation in various levels of abstraction, we have devised a method that, given a hierarchical ontology, generates a conceptual space of desired granularity.

For the experiments described in this paper, we chose Wikipedia category-directed graph as our ontology, as it provides a constantly- updated, wide and deep coverage of our knowledge, but any other knowledge graph can be used instead. Since the edges in the Wikipedia graph are not labeled, we performed an additional step of assigning a score to each edge, based on its similarity to its siblings, which we named *siblings* score (see Appendix A).

A major strength of the hierarchical representation of concepts is its multiple levels of abstraction. For our purpose, that means that we can request a concept space with a given level of granularity. Given a concept graph $G$, we can define $d(c)$, the depth of each concept (node) as the length of the shortest path from the root. We designate by $C^i = \{c \in C | d(c) = i\}$ as the set of all concepts with a depth of exactly $i$. For example, MATHE-MATICS and HEALTH are concepts from $C^1$, and

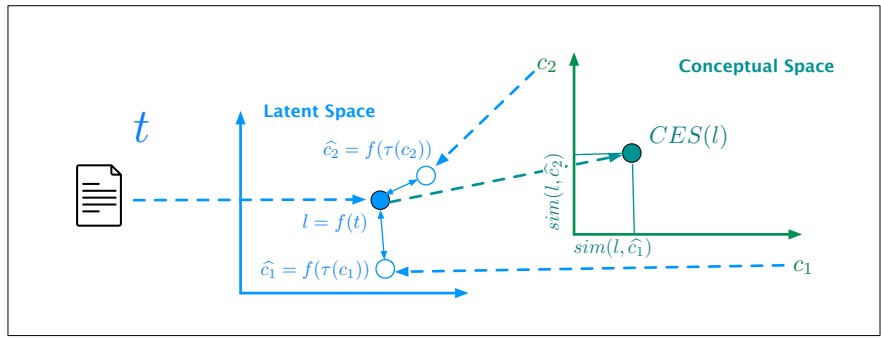

Figure 1: An outline of our methodology.

MATHEMATICAL TOOLS and PUBLIC HEALTH are their direct children and are concepts from $C^2$.

## 2.2 Selectively Refined Conceptual Spaces

One problem with fix-depth conceptual spaces is the large growth in the number of nodes with the increase in depth. For example, in our implementation, $|C^1| = 37$, $|C^2| = 706$ and $|C^3| = 3467$. Another problem arises in domain-specific tasks, where high-granularity concepts are needed in specific subjects but not in others. Lastly, it is often difficult to know ahead of time what is the required granularity for the given task.

We have therefore developed an algorithm that, given a contextual text $T' \subseteq T$ of input texts and the desired concept-space size, generates a concept space of that size with granularity tailored to $T'$. The main idea is to refine categories that are strongly associated with $T'$, thus enlarging the distances between the textual objects, allowing for more refined reasoning. We use the symbol $C^*$ to indicate a concept space that is created this way.

The algorithm (1) starts with $C^1$ as its initial concept space. It then iterates until the desired size is achieved. At each iteration, the contextual text $T'$ is embedded into the *current* space using CES. The concept with the largest weight after the projection to CES is then selected for expansion. The intuition is that this concept represents a main topic of the text, and will therefore benefit the most from a more refined representation. The algorithm selects its best $p\%$ children for some $p$, judged by their *siblings* score, and adds them to the current conceptual space. In addition, the algorithm utilizes a flag $removeP$ to decide whether to remove the expanded concept. We observed that retaining the parent can often improve the quality of the model interpretation.

If the embedding is used for a classification task,

---

**Algorithm 1** Selective Refinement
**Input:** $T', size, removeP$
**Output:** $C^*$
   $C \leftarrow C^1$
   **while** $|C| < size$ **do**
      $emb \leftarrow AVG_{t \in T'}\left(\text{CES}^{f,C,\tau}(f(t))\right)$
      $\hat{c} \leftarrow$ concept in $C$ with max weight in $emb$
      $best \leftarrow$ p% of $children(\hat{c})$ with highest *siblings* score
      $C \leftarrow C \cup best$
      **if** $removeP$ is True **then**
         $C \leftarrow C \setminus \hat{c}$
      **end if**
   **end while**
   **return** $C$

---

we can utilize the labels of the training examples alongside their text. We assign to each concept the set of examples for which it is the top concept. The entropy of this set is then combined linearly with the text-based weight described above to determine its final value. As before, the node with the maximal value is chosen for expansion. The underlying intuition is that concepts representing texts from different classes require refinement to allow a better separation.

## 2.3 Mapping Concepts to Text

The function $\tau$ maps concepts to text. When the concepts in the ontology have meaningful names, such as in the case of Wikipedia categories, we can just use $\tau$ that maps into these names. We have also devised a more complex function, $\hat{\tau}$, that maps a concept to a concatenation of the concept name with the names of its children[2]. Given a concept $c$ with name $t_c$ and children names $t_{c_1}$ and $t_{c_2}$, $\hat{\tau}(c) = "t_c \ such \ as \ t_{c_1} \ and \ t_{c_2}"$. This

---

[2] We take the two best children (the highest *siblings* score)

approach has two advantages: It exploits the elaborated knowledge embedded in the ontology for potentially more accurate mapping, and it produces full sentences, which may be a better fit for $f$ that was trained on sentences.

## 3 Empirical Evaluation

It is not easy to evaluate an algorithm whose task is to create an understandable representation that matches the original incomprehensible embedding. We performed a series of experiments, including a human study, that show that our method indeed achieved its desired goal. For all the experiments, we have used RoBERTa sentence embedding model[3] (Reimers and Gurevych, 2019; Liu et al., 2019) as our $f$, unless otherwise specified. All models used in this work were applied with their default parameters. Whenever the concept space $C^*$ was used, we set $size = 768$ to match the size used by SRoBERTa, but we observed that using much smaller values yielded almost as good results. The default value for $removeP$ is false. For $\tau$, the function that maps concepts to text, we have just used the text of the concept name (with a length of 4.25 words on average in $G$). [4]

### 3.1 Qualitative Evaluation

We first show several examples of conceptual representations created by CES to get some insight into the way that our method works. We have applied SRoBERTa to 3 sentences from 3 different recent CNN articles to get 3 latent embedding vectors. We have used the first 10 sentences of each article as the contextual text $T'$ for generating $C^*$.

Table 1 shows the conceptual embeddings generated by CES. We show only the 3 top concepts with their associated depth. Observe that the conceptual vectors are understandable and intuitively capture the semantics of the input texts. Note that the representations shown are not based on some new embedding method, but reflect SRoBERTa's understanding of the input text. In Appendix E, we study, using the same examples, the effect of the concept-space granularity on the conceptual representation, using a fixed-depth concept space instead of $C^*$. Lastly, in Appendix F, we study, using the

same examples, the difference in the representation of two additional models (SBERT and ST5).

### 3.2 Evaluation on Classification Tasks

To show that our representation matches the original one generated by the LLM, we first show that learning using the original embedding dimensions as features and learning using the conceptual features yield *similar* classifiers. Most works try to show such similarity by comparing accuracy results. This method, however, is prone to errors. Two classifiers might give us an accuracy of 80%, while agreeing only on 60% of the cases. Instead, we use a method that is used for *rater agreement*, reporting two numbers: the raw agreement and Cohen's kappa coefficient (Cohen, 1960).

We use the following datasets (all in English): AG News [5], Ohsumed and R8 [6], Yahoo (Zhang et al., 2015), BBC News (Greene and Cunningham, 2006), DBpedia 14 (Zhang et al., 2015) and 20Newsgroup [7]. We use only topical classification datasets, as the concept space we use does not include the necessary concepts needed for tasks like sentiment analysis. If a dataset has more than 10,000 examples, we randomly sample 10,000. The results are averaged over 10 folds. We use a random forest (RF) learning algorithm with 100 trees and a maximum depth of 5. The conceptual space used by CES is $C^*$, using the training set as the contextual text $T'$.

Table 2 shows the agreement between a random forest classifier trained on the LLM embedding and a classifier trained on the conceptual embedding generated by CES. For reference, we also show the agreement between the LLM-based classifier and a random classifier. We report raw agreement and kappa coefficient (with standard deviations). We can see that all the values are relatively high, indicating high agreement between the LLM embedding and CES's embedding. Note that Kappa can range from -1 to +1 with 0 indicating random chance. For the sake of completeness, we also report the accuracies of the two classifiers which are proved to be quite similar.

We repeated the experiment using a KNN classifier (n=5) with cosine similarity. The results are shown in Table 3. We can see much higher agree-

---

[3]Model all-distilroberta-v1 from Hugging Face. For simplicity we refer to it as SRoBERTa

[4]The total runtime for the experiments described here was 24 hours on 8 cores of Intel Xeon Gold 5220R CPU 2.20GHz. The graph creation from the full Wikipedia dump of 2020 took several days with a maximal memory allocation of 100GB.

[5]Available online:http://groups.di.unipi.it/~gulli/AG_corpus_of_news_articles.html

[6]Available online:https://www.kaggle.com/weipengfei/ohr8r52 used for Ohusmed and R8 datasets

[7]taken from sklearn datasets python library

| sentence | $c_1$ | $c_2$ | $c_3$ |
|---|---|---|---|
| This is now a very contagious virus | VIRUSES (3) | DISEASE OUTBREAKS (3) | VIRUS TAXONOMY (4) |
| The search for life on Mars and ocean worlds in our solar system | LIFE IN SPACE (2) | HYPOTHETICAL LIFE FORMS (2) | DISCOVERIES BY ASTRONOMER (3) |
| The bias in these AI systems presents a serious issue | ARTIFICIAL INTELLIGENCE (3) | MACHINE LEARNING (3) | COMPUTING AND SOCIETY (3) |

Table 1: Example of the model outputs on the sentences. The number in parenthesis is the depth of the concept.

ment between the LLM-based and CES-based classifiers.

We tested the sensitivity of our algorithm to the values of the $removeP = True$ and $\hat{\tau}$ parameters. The results are shown in Appendices B and C. We can see that both parameters have little effect on performance.

Appendix D includes additional positive results on the triplets dataset (Ein-Dor et al., 2018).

### 3.3 Evaluating Understandability

While these results look promising, they may not be sufficient to indicate that CES indeed reflects the semantics of the text according to the LLM. Consider the following hypothetical algorithm. Let $D$ be the size of the LLM embedding space. The algorithm selects $D$ random English words and assigns each to an arbitrary dimension. This hypothetical algorithm satisfies two requirements: Using it for the classification tasks will always be in 100% agreement with the original (as we merely renamed the features), and its generated representation will be understandable by humans, as we use words in natural language. However, it is clear that it does not convey to humans any knowledge regarding the LLM representation. In the next subsections, we describe a novel experimental design with humans and with other models. This design aims to validate our assertion that CES produces comprehensible representations that genuinely capture the semantics of the LLM embedding.

#### 3.3.1 Evaluation By Humans

We have designed a human experiment with the goal of testing the human understandability of the latent representation by observing only its conceptual mapping. The experiment tests the agreement, given a set of test examples, between two raters:

1. A classifier that was trained on a training set using the LLM embeddings.

2. A human rater that does not have access to the training set and does not have access to the test text. The only data presented to the human is the top 3 concepts of the CES representation of the LLM embedding. 3 graduate students were used for rating.

We claim that if there is a high agreement between the two, then the conceptual representation indeed reflects the meaning of the LLM embedding.

To allow classification by the human raters, out of the 7 datasets described in the previous subsection, we chose the 4 that have meaningful names for the classes. To make the classification task less complex for the raters, we randomly sampled two classes from each dataset, thus creating a binary classification problem. For each binary dataset, we set aside 20% of the examples for training a classifier based on the LLM embedding, using the same method and parameters as in the previous subsection. The resulting classifier was then applied to the remaining 80% of the dataset.

Out of this test set, we sample 10 examples on which the LLM-based classifier was right and 10 on which it was wrong[8]. This is the test set that is presented to the human raters. Each test case is represented by the 3 top concepts of the CES embedding, after applying feature selection on the full embedding to choose the top 20% concepts. As before, the conceptual space is $C^*$ with $size = 768$ and with the training set used as contextual text $T'$. The instruction to the human raters was: *"A document belongs to one of two classes. The document is described by the following 3 key phrases (topics): 1, 2, and 3. To which of the two classes do you think the document belongs to?".* The final human classification of a test example was computed by the majority voting of 3 raters. For the LLM-based classification, we used two learning algorithms. The first is Random Forest (RF) with the

---
[8]except for the Ohsumed dataset where only 7 wrong answers were found

| dataset | Rand/LLM raw agreement | CES/LLM raw agreement | CES/LLM kappa coefficient | CES accuracy | LLM accuracy | accuracy diff |
|---|---|---|---|---|---|---|
| 20 Newsgroup | 0.05 | $0.61 \pm 0.02$ | $0.58 \pm 0.02$ | 56.4 | 68.0 | 11.6 |
| AG News | 0.25 | $0.87 \pm 0.01$ | $0.83 \pm 0.02$ | 84.9 | 85.7 | 0.8 |
| DBpedia 14 | 0.07 | $0.85 \pm 0.01$ | $0.84 \pm 0.01$ | 87.0 | 88.4 | 1.4 |
| Ohsumed | 0.10 | $0.69 \pm 0.01$ | $0.58 \pm 0.01$ | 40.4 | 41.5 | 1.1 |
| Yahoo | 0.10 | $0.63 \pm 0.02$ | $0.59 \pm 0.02$ | 57.9 | 61.5 | 3.6 |
| R8 | 0.23 | $0.87 \pm 0.01$ | $0.76 \pm 0.02$ | 79.9 | 79.8 | -0.1 |
| BBC News | 0.19 | $0.96 \pm 0.01$ | $0.95 \pm 0.02$ | 95.8 | 97.0 | 1.2 |

Table 2: Agreement (raw and kappa) between LLM- and CES-based RF classifiers.

| dataset | Rand/LLM raw agreement | CES/LLM raw agreement | CES/LLM kappa coefficient | CES accuracy | LLM accuracy | accuracy diff |
|---|---|---|---|---|---|---|
| 20 Newsgroup | 0.05 | $0.82 \pm 0.02$ | $0.81 \pm 0.02$ | 78.9 | 86.9 | 8.0 |
| AG News | 0.25 | $0.92 \pm 0.01$ | $0.90 \pm 0.01$ | 87.9 | 89.7 | 1.8 |
| DBpedia 14 | 0.07 | $0.93 \pm 0.01$ | $0.92 \pm 0.01$ | 94.1 | 94.0 | -0.1 |
| Ohsumed | 0.10 | $0.75 \pm 0.02$ | $0.73 \pm 0.02$ | 65.2 | 72.4 | 7.2 |
| Yahoo | 0.10 | $0.75 \pm 0.01$ | $0.73 \pm 0.02$ | 65.2 | 69.2 | 4.0 |
| R8 | 0.23 | $0.92 \pm 0.01$ | $0.87 \pm 0.02$ | 89.4 | 93.6 | 4.2 |
| BBC News | 0.19 | $0.98 \pm 0.01$ | $0.97 \pm 0.01$ | 97.1 | 97.7 | 0.6 |

Table 3: Agreement (raw and kappa) between LLM- and CES-based KNN classifiers.

| dataset | raw agreement with RF | raw agreement with NC |
|---|---|---|
| AG News | 0.85 | 0.80 |
| BBC News | 0.80 | 0.75 |
| Ohsumed | 0.65 | 0.82 |
| Yahoo | 0.65 | 0.75 |

Table 4: Human-RF and human-NC raw agreement.

same parameters as in Section 3.2. The second is Nearest-Centroid Classifier (NC) which computes the centroid of each class and returns the one closest to the test case.

Table 4 shows the raw agreement between the LLM-based and the human classification, for the two learning algorithms. Kappa coefficient was not computed as the test set is too small. The results are encouraging as they show quite a high agreement. Note that the learning algorithm had access to the full training set, while the human could see the conceptual representation of only the test case. Indeed, we can see that the agreement with the less sophisticated NC classifier is higher on average than the agreement with the RF classifier.

### 3.3.2 Evaluation by Other Models

We repeated the experiments of the last subsection, with the same test sets, but instead of using human raters, we used a LLM rater. The LLM rater receives the top 3 concepts, just like the human raters, and makes a decision by computing cosine similarity between its embedding of each class name to its embedding of the textual representation of the 3 concepts. The 3 LLMs used for rating are SBERT (Reimers and Gurevych, 2019) [9], ST5 (Ni et al., 2022) [10] and SRoBERTa. Note that the two uses of SRoBERTa are quite different. The one used for the original classification is based on a training set and a learning algorithm, while the model used for rating just computes similarity between the class name and the 3 concepts.

An alternative approach to ours is to assign a meaning to each dimension of the latent space. We denote this approach by Dimension Meaning Assignment (DMA). We have designed two competitors that represent the DMA approach.

The first one, termed $DMA^{words}$, is based on a

---

[9] Model bert-base-nli-mean-tokens from Hugging Face
[10] Model sentence-t5-large from Hugging Face

vocabulary of 10K frequent words [11]. We represent each word by our LLM, yielding 10K vectors of size 768. We now map each dimension to the word with the highest weight for it. We make sure that the mapping is unique. The second one, which we call $DMA^{concepts}$, is built in the same way, using, instead of words, the concepts in $C^3$. Lastly, $DMA^{C^*}$ is added as an ablation experiment where the transformation part of our method is turned off. Table 5 shows the results expressed in raw agreement. We can see that CES method performs better than the alternatives (except for two test cases).

The previous two subsections (3.3.1 and 3.3.2) have presented evidence supporting our fundamental claim that the conceptual representation generated by CES accurately captures the semantic content of the input text based on the LLM model.

# 4 CES Application

## 4.1 Using CES for Comparing Models

One major feature of our methodology is that it allows us to gain an understanding of the semantics of trained models. This allows us when considering alternative models, to compare their semantics, to understand the differences between their views of the world, and compare their potential knowledge gaps. We demonstrate this by comparing the views of three LLMs, SBERT, ST5, and SRoBERTa on two example texts, by observing their conceptual representations in $C^3$ generated by CES.

Table 6 shows the top 3 concepts of the vector generated by CES for the 3 LLMs given the text "FC Barcelona". We can see that while SRoBERTa and ST5 give high weight to the sport aspect of the input text, SBERT does not.

To validate this observation, we compare, for each of the 3 models, the cosine similarity in the latent space between "FC Barcelona" and the sport-related phrase "Miami Dolphin", to its similarity to the city-related phrase "Politics in Spain".

The results support our observation. SBERT embedding is more similar to the city aspect embedding while the two others are more similar to the sports text embedding.

In Table 7, for the input "Manhattan Project", we can see that ST5 gives high weight to the military project while SBERT gives high weight to concepts related to New York and to theater. SRoBERTa recognizes both aspects.

## 4.2 Using CES for LLM Tracing

Another application of CES is analyzing the layers of the LLM, in a similar way to the Logit lens method (nostalgebraist, 2020). This can be very useful for debugging the model. We show here an example of tracing the changes of the embedding through the layers of BERT and GPT2[12].

We create a representation for each layer by calculating the average of the token embeddings within that layer[13]. We then use CES to map these vectors to the conceptual space. We then trace the relative weight of each concept throughout the layers to gain an understanding of the modeling process.

In this case study, we analyze the text "Government" using $C^3$ conceptual space. We follow 6 concepts: the 3 top ones for the initial layer and the 3 top ones for the final layer. Figure 2 shows the changes in the weights of the concepts throughout the modeling process. The y axis shows the ranking of each concept.

The figure offers a clear visualization of the changes in the relative weights of these concepts across the different layers. Notably, in Figure 2a, the concepts TRANSPORT, MEDICINE, and CORRUPTION, which had low rankings in the initial layer, have significantly ascended to become the top concepts in the final layer. A similar transition using different concepts is found in Figure 2b.

# 5 Related Work

The problem of interpretability has received significant attention in recent years. A large body of research (Ribeiro et al., 2016a; Lundberg and Lee, 2017b; Yeh et al., 2020; Rajani et al., 2020; Ribeiro et al., 2018; Ebrahimi et al., 2018; Ross et al., 2021; Wu et al., 2021) is devoted to generate an explanation for the *decision* of the model (mostly classification). Many methods utilize nearby examples or counterfactuals to provide users with reasoning behind the decision.

Several works set a goal, like ours, of understanding the model itself, rather than its decisions. Most of these works attempt to assign some meaning to the dimensions, either of the original latent space or of a different space that the original one is transformed to.

---

[11]https://www.mit.edu/ ecprice/wordlist.10000

[12]Model bert-base-uncased and gpt2 from Hugging Face, including the input initial embedding.

[13]Other methods, such as using the last token, can be easily incorporated.

| Evaluation Model | Method | Yahoo | BBC | AG News | Ohsumed |
|---|---|---|---|---|---|
| SBERT | $DMA^{words}$ | 0.60 / 0.60 | 0.55 / 0.80 | 0.55 / 0.50 | 0.65 / 0.47 |
| | $DMA^{concepts}$ | 0.65 / 0.55 | 0.55 / 0.70 | 0.50 / 0.35 | 0.65 / 0.47 |
| | $DMA^{C^*}$ | 0.60 / 0.50 | 0.60 / 0.75 | 0.65 / **0.70** | 0.59 / 0.42 |
| | CES | **0.80 / 0.90** | **0.70 / 0.85** | **0.75** / 0.60 | **0.71 / 0.53** |
| ST5 | $DMA^{words}$ | 0.65 / 0.65 | 0.35 / 0.60 | 0.45 / 0.50 | 0.65 / 0.47 |
| | $DMA^{concepts}$ | 0.70 / 0.70 | 0.45 / 0.30 | 0.55 / **0.60** | 0.53 / 0.35 |
| | $DMA^{C^*}$ | 0.60 / 0.60 | 0.65 / 0.40 | 0.55 / **0.60** | 0.53 / 0.35 |
| | CES | **0.80 / 0.90** | **0.80 / 0.75** | **0.60** / 0.55 | **0.76 / 0.82** |
| SRoBERTa | $DMA^{words}$ | 0.50 / 0.70 | 0.35 / 0.40 | 0.55 / 0.60 | 0.59 / 0.41 |
| | $DMA^{concepts}$ | 0.60 / 0.40 | 0.35 / 0.50 | 0.60 / 0.45 | 0.71 / 0.53 |
| | $DMA^{C^*}$ | 0.40 / 0.60 | 0.55 / 0.40 | 0.35 / 0.40 | 0.71 / 0.53 |
| | CES | **0.85 / 0.85** | **0.75 / 0.80** | **0.70 / 0.65** | **0.82 / 0.76** |

Table 5: Evaluation by SRoBERTa-RF / SRoBERTa-NC.

| Model | $c_1$ | $c_2$ | $c_3$ | d(t,"Miami Dolphins") | d(t,"politics in Spain") |
|---|---|---|---|---|---|
| SBERT | GOVERNMENT OF SPAIN | SPANISH PEOPLE | CATALAN CULTURE | 0.42 | 0.57 |
| SRoBERTa | TEAMS | SPORT BY CITY | SAINTS | 0.40 | 0.30 |
| ST5 | TEAM SPORTS | SPORTS TEAMS | PEOPLE IN SPORTS BY ORGANIZATION | 0.79 | 0.75 |

Table 6: t="FC Barcelona", FC Barcelona top 3 concepts using CES and validation by the LLM.

| Model | $c_1$ | $c_2$ | $c_3$ | d(t,"Nuclear bomb") | d(t,"New York") |
|---|---|---|---|---|---|
| SBERT | CITY-STATES | NEW YORK CITY NIGHTLIFE | THEATRE BY CITY | 0.49 | 0.74 |
| SRoBERTa | MILITARY PROJECTS | NEW YORK CITY NIGHTLIFE | SPACE PROGRAMS | 0.36 | 0.34 |
| ST5 | NUCLEAR TECHNOLOGY | NUCLEAR POWER | NUCLEAR ENERGY | 0.84 | 0.79 |

Table 7: t="Manhattan Project", Manhattan Project top 3 concepts using CES and validation by the LLM.

One relatively early approach tries to find orthogonal or close to orthogonal transformations of the original embedding matrix (Dufter and Schütze, 2019; Park et al., 2017; Rothe and Schütze, 2016) such that a set of words with high weight in a given dimension are related and thus hopefully represent some significant concept. The advantage of these orthogonal methods is that they do not lose information due to the orthogonality. Several of these works (Arora et al., 2018; Murphy et al., 2012; Subramanian et al., 2018; Ficsor and Berend, 2021; Berend, 2020) transform the original embedding to a sparse one to improve the interpretability of each dimension. One limitation of these methods is their reliance on an embedding/dictionary matrix.

Senel et al. (2018) assigns a specific concept to each dimension. Note that our work is different as it does not assume that each latent dimension corresponds to a human-understandable concept.

Recent methods (Dar et al., 2022; nostalgebraist, 2020) assume access to the model's internals, particularly the un-embedding matrix, to map a latent vector to the token space.

Other works (Yun et al., 2021; Molino et al., 2019) created new tools to help interpret the model. Yun et al. (2021) uses dictionary learning to view the transformer model as a linear superposition of transformer factors. Molino et al. (2019) introduces a tool for doing simple operations such as PCA and t-SNE on embedding.

Probing methods try to interpret the model by studying its internal components. Vig et al. (2020)

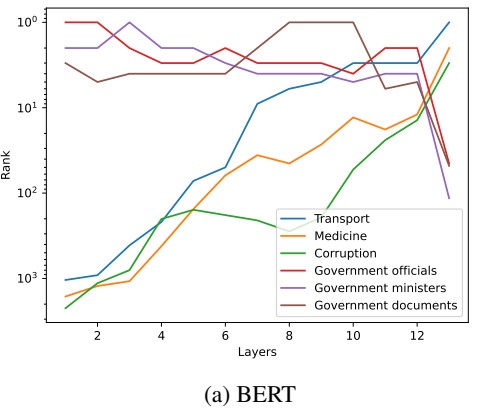

(a) BERT

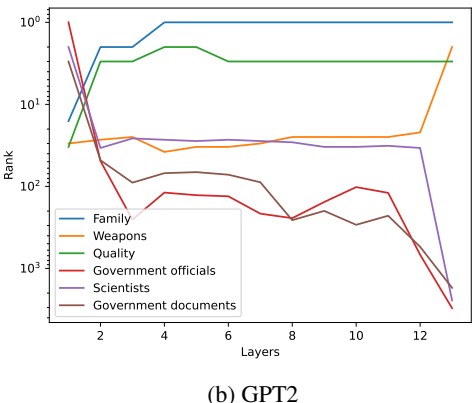

(b) GPT2

Figure 2: BERT/GPT2 layers for 'Government' text.

make changes to the input to find out what parts of the model (specific attention heads) a bias comes from. Tenney et al. (2019) use probing on BERT model to find the role of each layer in the text interpretation process. Bau et al. (2019) and Dalvi et al. (2019) show how linguistic properties are distributed in the model and in specific neurons. Clark et al. (2019) create an attention-based probing classifier to find out what information is captured by each attention head of BERT. Lastly, Sommerauer and Fokkens (2018) use supervised classifiers to extract semantic features.

Some works (Mathew et al., 2020; An et al., 2022; Bouraoui et al., 2022; Faruqui et al., 2015; Senel et al., 2022; Şenel et al., 2021) tackle the problem by training or retraining to create a new interpretable model. Unlike those methods, our approach focuses on understanding the original models while preserving their performance, rather than using interpretable models as substitutes.

## 6 Conclusion

In this work, we introduce a novel approach to LLM interpretation that maps the latent embedding space into a space of concepts that are well-understood by humans and provide good coverage of the human knowledge. We also present a method for generating such a conceptual space with an on-demand level of granularity.

We evaluate our method by an extensive set of experiments including a novel method for evaluating the correspondence of the conceptual embedding to the *meaning* of the original embedding both by humans and by other models. Finally, we showed applications of our method for comparing models, analyzing the layers of the model, and debugging.

## 7 Limitations

There are several limitations to the work presented here:

1. For the tracing application (Section 4.2), we used a rather limited (but common) approach of averaging the embedding vectors of each token.

2. Most of our experiments were performed using only the SRoBERTa model.

3. We did not include experiments using CES for explanation and for debugging. Such application will be performed in future work

4. Our evaluation was done using only Wikipedia category graph as an ontology. Using alternative knowledge graphs can be of interest.

## 8 Ethics Statement

The primary objective of our method is to facilitate a deeper comprehension of the embedding space. Our model serves as a tool to enhance understanding of the underlying model. By utilizing the model, offensive mappings in the concept space of CES can be revealed. However, it is important to note that our model is strictly intended for the purpose of assisting in understanding and debugging problems in LLMs.

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

## A  *Siblings* score

Let $G = (V, E)$ be a knowledge graph, where $V$ is a set of concepts and $E \subseteq V \times V$ is a set of links between concepts. Let $Obj(c)$ be the set of objects belonging to concept $c$. We say that $c_1$ *is-a* $c_2$ if $Obj(c_1) \subseteq Obj(c_2)$. We define $parents(c) = \{c' \in V | (c', c) \in E\}$ and $children(c) = \{c' \in V | (c, c') \in E\}$. Given a node $c$ and a parent node $p$, we define $siblings(c, p) = children(p) - \{c\}$.

The main idea behind our method of detecting *is-a* links is that a set of siblings connected to a specific parent through *is-a* links should be similar. We estimate the similarity between a node and its siblings by the similarity between their set of parents. Instead of using a binary decision, we chose to assign a continuous value in $[0, 1]$ that will be used by our algorithms for generating conceptual spaces.

We can now define the *siblings* score of an edge $(p, c)$ as:

$$AVERAGE_{s \in siblings(c,p)} \frac{|parents(c) \cap parents(s)|}{|parents(c)|}$$

We remove from each node $\lambda\%$ (35% in our experiments) of its parent links with the lowest *siblings* score.

## B  Testing the effect of the $removeP$ parameter

Our algorithm for generating on-demand conceptual spaces 2.2 retains a parent after expanding it and adding its children. This has several advantages, but we commonly prefer embedding spaces that are orthogonal. In this section, we test the performance of our method if we delete the parent after expansion (controlled by the $removeP$ parameter). We ran the classification task as described in Section 3.2 with the only difference that the $removeP$ parameter is set to True. The results are shown in Table 8. We can see that the differences are insignificant.

| dataset | raw agreement | | kappa coef | | raw agreement removeP True | | kappa coef removeP True | |
|---|---|---|---|---|---|---|---|---|
| 20 News-group | 0.61 | ± 0.02 | 0.58 | ± 0.02 | 0.62 | ± 0.02 | 0.59 | ± 0.02 |
| AG News | 0.87 | ± 0.01 | 0.83 | ± 0.02 | 0.87 | ± 0.01 | 0.83 | ± 0.02 |
| DBpedia 14 | 0.85 | ± 0.01 | 0.84 | ± 0.01 | 0.87 | ± 0.01 | 0.86 | ± 0.01 |
| Ohsumed | 0.69 | ± 0.01 | 0.58 | ± 0.01 | 0.71 | ± 0.02 | 0.59 | ± 0.02 |
| Yahoo | 0.63 | ± 0.02 | 0.59 | ± 0.02 | 0.64 | ± 0.01 | 0.60 | ± 0.01 |
| R8 | 0.87 | ± 0.01 | 0.76 | ± 0.02 | 0.88 | ± 0.01 | 0.76 | ± 0.02 |
| BBC News | 0.96 | ± 0.01 | 0.95 | ± 0.02 | 0.96 | ± 0.02 | 0.95 | ± 0.02 |

Table 8: LLM- and CES-based classifiers' agreement. **Using $removeP$ as True.**

| dataset | raw agreement | | kappa coef | | raw agreement $\widehat{\tau}$ | | kappa coef $\widehat{\tau}$ | |
|---|---|---|---|---|---|---|---|---|
| 20 News-group | 0.61 | ± 0.02 | 0.58 | ± 0.02 | 0.62 | ± 0.01 | 0.59 | ± 0.02 |
| AG News | 0.87 | ± 0.01 | 0.83 | ± 0.02 | 0.87 | ± 0.01 | 0.83 | ± 0.01 |
| DBpedia 14 | 0.85 | ± 0.01 | 0.84 | ± 0.01 | 0.84 | ± 0.01 | 0.83 | ± 0.02 |
| Ohsumed | 0.69 | ± 0.01 | 0.58 | ± 0.01 | 0.69 | ± 0.01 | 0.58 | ± 0.02 |
| Yahoo | 0.63 | ± 0.02 | 0.59 | ± 0.02 | 0.64 | ± 0.01 | 0.59 | ± 0.02 |
| R8 | 0.87 | ± 0.01 | 0.76 | ± 0.02 | 0.88 | ± 0.01 | 0.77 | ± 0.02 |
| BBC News | 0.96 | ± 0.01 | 0.95 | ± 0.02 | 0.96 | ± 0.01 | 0.95 | ± 0.02 |

Table 9: LLM- and CES-based classifiers' agreement. **Using $\widehat{\tau}$ function.**

## C Testing the effect of the $\tau$ function

One of the major components of our method is the $\tau$ function that maps a concept into a text object that is then converted to a latent vector. For the experiments described in this work, we have used $\tau$ that just outputs the concept names. In this section, we repeat the classification tests with $\widehat{\tau}$ (see Section 2.3). Table 9 shows the results. We can see that the differences are insignificant.

| models | raw agreement | kappa coef |
|---|---|---|
| True labels and LLM labels | 0.726 | 0.452 |
| True labels and $C^3$ labels | 0.692 | 0.384 |
| LLM labels and $C^3$ labels | **0.820** | **0.640** |

Table 10: Raw agreement and kappa coefficient between SRoBERTa LLM labels, True labels and CES using $C^3$ labels on Wikipedia triplet dataset.

## D Evaluation on a Similarity Task

In this section, we use the conceptual representation in the context of an algorithm that estimates semantic similarity between sentences by measuring cosine similarity between their embeddings. Specifically, we evaluate the agreement between using the latent embedding generated by SRoBERTa and using the conceptual embedding generated by CES with $C^3$ (We cannot use $C^*$ since we do not have any contextual text to be used as $T'$).

The dataset used is the triplet test that was generated from Wikipedia articles (Ein-Dor et al., 2018). Each test consists of three sentences, all from the same Wikipedia article. Two sentences are from the same section and the third is from a different section. A sentence is labeled as more similar to the one from the same section than to the one from the other section. We used a subset of 1000 triplets randomly sampled from the full dataset.

The results are shown in Table 10. We can see that CES embedding and SRoBERTa embedding have a high raw agreement and Kappa coefficient, larger than their agreement with the true label.

## E A Qualitative Evaluation using Fixed-Depth Concept Spaces

We ran the same qualitative evaluation, as shown in Section 3.1, on the sentences taken from CNN. Instead of using the concept space $C^*$, we used fixed-depth spaces, $C^1$, $C^2$, and $C^3$. Our goal is to study the effect of the granularity of the concept space on the way the latent vectors are represented.

The top five concepts of each input sentence for each concept space are presented in Tables 11, 12 and 13. For comparison, we also include the top concepts of the $C^*$ concept space. We can see the refinement of the top concepts as the depth grows. Using $C^1$, the conceptual representation gives a very general and non-specific account of the text's meaning. Using the more refined $C^2$ and $C^3$

concept spaces, we can gain a deeper understanding of the input text. We can also notice that $C^*$ has an advantage over the fixed-depth alternatives as it can use more refined concepts when needed without compromising the size.

## F    A Qualitative Evaluation using Different Models

We ran the same qualitative evaluation, as shown in Section 3.1, on the sentences taken from CNN on all three models: SBERT, ST5, and SRoBERTa. Our goal is to study the difference between the models in a qualitative test.

The top five concepts of each input sentence for each model are presented in Tables 14, 15 and 16. It seems that all of the models "understood" the texts similarly. In Table 16 we can see a difference between SBERT and the other models. It seems that SBERT gave more weight to the word *bias* while the other models gave more weight to the word *AI* from the input sentence.

| $c$ | $C^1$ | $C^2$ | $C^3$ | $C^*$ |
|---|---|---|---|---|
| $c_1$ | MASS MEDIA | ORGANIZATIONS ASSOCIATED WITH THE COVID-19 PANDEMIC | VIRUSES | VIRUSES (3) |
| $c_2$ | PEOPLE | GLOBAL HEALTH | INFECTIOUS DISEASES | DISEASE OUTBREAKS (3) |
| $c_3$ | HEALTH | HEALTH DISASTERS | DISEASE OUTBREAKS | VIRUS TAXONOMY (4) |
| $c_4$ | CULTURE | REPRODUCTION | VACCINATION | COVID-19 PANDEMIC IN EUROPE (5) |
| $c_5$ | WORLD | EVOLUTION | VIRAL MARKETING | COVID-19 PANDEMIC IN ASIA (5) |

Table 11: An example of the top concepts of the model's output for the input **"This is now a very contagious virus"**, taken from CNN.

| $c$ | $C^1$ | $C^2$ | $C^3$ | $C^*$ |
|---|---|---|---|---|
| $c_1$ | LIFE | LIFE IN SPACE | EXTRATERRESTRIAL LIFE | LIFE IN SPACE (2) |
| $c_2$ | WORLD | HYPOTHETICAL LIFE FORMS | MESOZOIC LIFE | HYPOTHETICAL LIFE FORMS (2) |
| $c_3$ | SCIENCE AND TECHNOLOGY | ORIGIN OF LIFE | PALEOZOIC LIFE | DISCOVERIES BY ASTRONOMER (3) |
| $c_4$ | GEOGRAPHY | COSMOLOGY | EXPLORERS | ARTIFICIAL LIFE (2) |
| $c_5$ | HUMANITIES | FICTIONAL LIFE FORMS | POLAR EXPLORATION | ASTRONOMICAL CATALOGUES (2) |

Table 12: An example of the top concepts of the model's output for the input **"The search for life on Mars and ocean worlds in our solar system"**, taken from CNN.

| $c$ | $C^1$ | $C^2$ | $C^3$ | $C^*$ |
|---|---|---|---|---|
| $c_1$ | CONCEPTS | INTELLECTUAL COMPETITIONS | ARTIFICIAL INTELLIGENCE | ARTIFICIAL INTELLIGENCE (3) |
| $c_2$ | POLICY | LEARNING | COLLECTIVE INTELLIGENCE | MACHINE LEARNING (3) |
| $c_3$ | ETHICS | ISSUES IN ETHICS | COMPUTER ETHICS | COMPUTING AND SOCIETY (3) |
| $c_4$ | POLITICS | SOCIAL SYSTEMS | MACHINE LEARNING | INTELLECTUAL COMPETITIONS (2) |
| $c_5$ | SCIENCE AND TECHNOLOGY | CONCEPTUAL SYSTEMS | CLASSIFICATION SYSTEMS | INFORMATION SYSTEMS (3) |

Table 13: An example of the top concepts of the model's output for the input **"The bias in these AI systems presents a serious issue"**, taken from CNN.

| $c$ | SBERT | ST5 | SRoBERTa |
|---|---|---|---|
| $c_1$ | DISEASE OUTBREAKS (3) | VIRUSES (3) | VIRUSES (3) |
| $c_2$ | DISASTERS (2) | COVID-19 PANDEMIC IN EUROPE (5) | DISEASE OUTBREAKS (3) |
| $c_3$ | DOOMSDAY SCENARIOS (3) | COVID-19 PANDEMIC IN ASIA (5) | VIRUS TAXONOMY (4) |
| $c_4$ | HAZARDS (3) | DISEASE OUTBREAKS (3) | COVID-19 PANDEMIC IN EUROPE (5) |
| $c_5$ | CRIMINAL PROCEDURE (4) | PUBLIC HEALTH EMERGENCY OF INTERNATIONAL CONCERN (3) | COVID-19 PANDEMIC IN ASIA (5) |

Table 14: An example of the top concepts of the model's output for the input **"This is now a very contagious virus"**, taken from CNN. The number in parenthesis is the depth of the concept in the concept graph.

| $c$ | SBERT | ST5 | SRoBERTa |
|---|---|---|---|
| $c_1$ | ATMOSPHERE OF EARTH (3) | LIFE IN SPACE (2) | LIFE IN SPACE (2) |
| $c_2$ | OUTER SPACE (3) | DISCOVERIES BY ASTRONOMER (3) | HYPOTHETICAL LIFE FORMS (2) |
| $c_3$ | SOLAR SYSTEM IN FICTION (4) | HUMAN SPACEFLIGHT (3) | DISCOVERIES BY ASTRONOMER (3) |
| $c_4$ | DISCOVERIES BY ASTRONOMER (3) | ASTROBIOLOGY (3) | ARTIFICIAL LIFE (2) |
| $c_5$ | ASTRONOMICAL LOCATIONS IN FICTION (4) | ASTRONOMICAL OBJECTS (3) | ASTRONOMICAL CATALOGUES (4) |

Table 15: An example of the top concepts of the model's output for the input **"The search for life on Mars and ocean worlds in our solar system"**, taken from CNN. The number in parenthesis is the depth of the concept in the concept graph.

| $c$ | SBERT | ST5 | SRoBERTa |
|---|---|---|---|
| $c_1$ | CONFLICTS (2) | ARTIFICIAL NEURAL NETWORKS (4) | ARTIFICIAL INTELLIGENCE (3) |
| $c_2$ | SEXUALITY AND GENDER-RELATED PREJUDICES (3) | MACHINE LEARNING (3) | MACHINE LEARNING (3) |
| $c_3$ | GLOBAL CONFLICTS (3) | ARTIFICIAL INTELLIGENCE (3) | COMPUTING AND SOCIETY (3) |
| $c_4$ | POLITICAL CORRUPTION (2) | SOCIAL SYSTEMS (2) | INTELLECTUAL COMPETITIONS (2) |
| $c_5$ | ANTI-ISLAM SENTIMENT (4) | ARTIFICIAL LIFE (2) | INFORMATION SYSTEMS (3) |

Table 16: An example of the top concepts of the model's output for the input **"The bias in these AI systems presents a serious issue"**, taken from CNN. The number in parenthesis is the depth of the concept in the concept graph.