# OpenReview forum: "Interpreting Embedding Spaces by Conceptualization"
_EMNLP/2023/Conference — EMNLP 2023 Main_

### Official Review · Reviewer_nPyK · 2023-08-04

**Soundness:** 3

**Excitement:**

4: Strong: This paper deepens the understanding of some phenomenon or lowers the barriers to an existing research direction.

**Paper Topic And Main Contributions:**

In this article, authors propose to project the representation of a piece of text obtained with a deep language model into a “concept” space (more precisely, computing cosine similarity between the piece of text representation and each concept representation) for improving LLM interpretability. In the experiments, these concepts are built using Wikipedia categories, for which selection depends on the required level of granularity. The method is evaluated on  7 common datasets with both human and automatic evaluations.

**Questions For The Authors:**

Concepts representations are not contextualized. What about ambiguous context? The method proposed in 2.3 might circumvent this issue, but it should be properly tested.
This approach is related with topic modeling, in the specific case of fixed topic. How this approach competes against these supervised topic modeling approaches ?
If there exist some correlation in the concept space (concepts are not orthogonal enough in term of semantic, e.g. in the case of colinear concept's representations) then a lot of the initial information will be lost (contrary to what is stated in 3.3). The approach seems not to lose that much on classification task, but how the chosen concepts impact the performance of the obtained representation should be carefully evaluated.


**Reasons To Accept:**

Simple yet effective method
The approach is carefully evaluated, both with human in the loop and automatic/semi-automatic protocols.
The method can be adapted to many application domains, as one can tailor the concept space to the dataset at hand.


**Reasons To Reject:**

The article lacks a comparison with competitors, that are well presented on the related work section.
The article lacks an analysis of the impact of the chosen concepts (even on the granularity).


**Reproducibility:**

2: Would be hard pressed to reproduce the results. The contribution depends on data that are simply not available outside the author's institution or consortium; not enough details are provided.

**Reviewer Confidence:**

3: Pretty sure, but there's a chance I missed something. Although I have a good feel for this area in general, I did not carefully check the paper's details, e.g., the math, experimental design, or novelty.

---

> ### Author Rebuttal · Authors · 2023-08-28
>
> We first want to thank you for the review.
>
> Following the questions in the reviews, we performed the experiments of Section 3.2 with a simpler classifier, KNN (K=5, DIstance function is cosine).  The results, which are much better than the RF ones, are shown here.  We will add this table to the paper.
>
> | Dataset | raw agreement| kappa coef| CES acc| LLM acc|
> |---------|--------------|-----------|--------|--------|
> | 20 news | 0.82         | 0.81      | 79     | 87     |
> | ag_news | 0.92         | 0.90      | 88     | 90     |
> | DBpedia | 0.93         | 0.92      | 94     | 94     |
> | Ohsumed | 0.75         | 0.73      | 65     | 72     |
> | Yahoo   | 0.75         | 0.73      | 65     | 69     |
> | R8      | 0.92         | 0.87      | 90     | 94     |
> | BBC     | 0.98         | 0.97      | 97     | 98     |
>
> Regarding the questions you have:
>
> - The article lacks a comparison with competitors
>   - Many of the existing approaches that create new representation attempt to transform the LLM representation to a more interpretable one and focus on showing that the new representation leads to a good or better performance.  The goal of our work is completely different.  Our goal is to convey to a human user the semantics of any given LLM representation (rather than altering it).  In order to compare to other approaches with respect to this goal, we would need to use each of the methods with a black-box LLM, generate some human-understandable semantic representation of each input sentence, test the correlation between this representation and the original latent representation, and test the comprehensibility to humans.  We believe that the work presented here is one of the first to provide such a comprehensive evaluation.  We found it extremely difficult to compare CES to alternatives using such a methodology but we hope that others working towards similar goals will build and expand the methodology introduced here.
> - The article lacks an analysis of the impact of the chosen concepts (even on the granularity)
>   - In Appendix F we show a qualitative evaluation of different granularity of fixed and flexible depth of conceptual space on the embedding. In those results, we can see the advantage of the flexible-based conceptual space over the fixed one.
>
> - Concepts representations are not contextualized. What about ambiguous context? The method proposed in 2.3 might circumvent this issue, but it should be properly tested.
>   - The average length of a concept name is 4.25 which is sufficient in most cases.  However, we included an additional type of \tau, called \widehat\tau (section 2.3) that adds children's concept names as a context. An experiment comparing the two can be found in Appendix D. In the future, we consider adding some text from the Wikipedia articles belonging to the concept as context.
> - This approach is related with topic modeling, in the specific case of fixed topic. How this approach competes against these supervised topic modeling approaches ?
>   - As we understand, topic modeling uses unsupervised ML to identify clusters or groups of similar words within a text to create a topic. Our work has one step of generating the conceptual space given a context and another step of projecting the embedding space into the conceptual one. Our work can be used for explaining LLM-based topic modeling.
> - If there exist some correlation in the concept space (concepts are not orthogonal enough in term of semantic, e.g. in the case of colinear concept's representations) then a lot of the initial information will be lost (contrary to what is stated in 3.3)
>   - We believe that orthogonality is less important for generating explanation, however,  to make the concept space “more orthogonal”,  we have a parameter removeP, If this parameter is True the algorithm in 2.2 removes the parent node of a node that is added to the concept space, this modification should help with the orthogonality. We ran a test with this parameter in Appendix C.

---

### Official Review · Reviewer_EU1c · 2023-08-04

**Soundness:** 4

**Excitement:**

4: Strong: This paper deepens the understanding of some phenomenon or lowers the barriers to an existing research direction.

**Paper Topic And Main Contributions:**

The paper proposes an approach for transforming the otherwise opaque dimensions of contextualized sentence embeddings into directly interpretable categories of an ontology and evaluate them in various topic classification tasks.
The proposed approach (dubbed CES) turns some concept inventory into text, transforms them into the representation space of a sentence encoder, finally constructs a transformation matrix of these vectorized concept vectors that maps the latent vectorial representations of the encoder into such a space in which the individual dimensions are directly tied to one of the concepts from the concept inventory.
The idea is simple and sound, even though the concepts seem to be converted into the representation space somewhat naively.

The representations with interpretable dimensions are evaluated in 7 different document classification tasks.
One limitation of the work is that it only works for document classification.

The quality of the transformed representations is assessed is via measuring the prediction similarity of such random forest classifiers that use either the transformed or the original representations as document features.
As the goal of the transformation was to create interpretable representations, using a more transluent classifier (other than a random forest with 100 classifiers) would be more adequate.
Could the prediction similarity be due to the large number of classifiers used in the ensemble of classifiers?

Besides the agreement rate of the classifiers relying on the original and the interpretable features, their accuracy is equally important, which results are delegated to the appendix.
It seems that the use of the transformed representations degrade classification performance noticeably.
Some amount of performance drop is fine in exchange for more interpretable representations, but this kind of tradeoff should be made explicit and clearly articulated.

The understandability of the representations from the perspective of document classification is assessed by humans as well as in an automated way.
Related to the automated evaluation of the understandability of the transformed representations, the alternative baseline approaches that assign meaning to each dimension of the original latent space are too naive, i.e., the proposed CES approach has access to C*, the concept inventory tailored for the given task, whereas the alternative approach uses either a vocabulary of the most frequent words or a concept inventory that does not enjoy the benefit of being adjusted to the domain in question.
This way, it remains unknown to what extent CES performs better due to the more useful vocabulary or the actual approach itself.
When performing the automated undestandability assessment, based on the figures in Table 4, only 20 documents per dataset was used.
Since the automated assessment is not constrained by human labor, this kind of comparison might have been performed over more documents.

**Reasons To Accept:**

Developing interpretable (document) representations is important topic, for which a simple approach is provided.

**Reasons To Reject:**

The proposed approach only works for document classification, the applied baseline is too simple and the comparison with other existing approaches is lacking.
The classification performance drops quite noticeably compared to the use of the original (yet uninterpretable) representations.

**Reproducibility:**

4: Could mostly reproduce the results, but there may be some variation because of sample variance or minor variations in their interpretation of the protocol or method.

**Reviewer Confidence:**

5: Positive that my evaluation is correct. I read the paper very carefully and I am very familiar with related work.

**Typos Grammar Style And Presentation Improvements:**

The statement in the related work section that sparsification only works for static embeddings is not true, see e.g. [1,2,3].
[3], being an extension of (Senel et al., 2018) for the contextual case could serve as a stronger baseline as it is also capable of assigning human interpretable labels to the dimensions of transformed latent representations.

[1] Berend, Gábor. ["Sparsity makes sense: Word sense disambiguation using sparse contextualized word representations."](https://aclanthology.org/2020.emnlp-main.683/)
[2] Yun, Zeyu, et al. ["Transformer visualization via dictionary learning: contextualized embedding as a linear superposition of transformer factors."](https://aclanthology.org/2021.deelio-1.1/)
[3] Ficsor, Tamás, and Gábor Berend. ["Changing the Basis of Contextual Representations with Explicit Semantics." ](https://aclanthology.org/2021.acl-srw.25/)

---

> ### Author Rebuttal · Authors · 2023-08-28
>
> Thank you for the feedback as well as the points on other related work.
>
> Regarding the question you have:
>
> - One limitation of the work is that it only works for document classification.
>   - It is important to understand that the classification experiments described in this paper have one goal: to show that the human-understandable conceptual representation and the original latent representation are strongly correlated.  We do not propose to use the conceptual features for classification - instead, we propose to use the conceptual features for explanation, debugging, and debiasing as shown in Section 4.
> - As the goal of the transformation was to create interpretable representations, using a more transluent classifier (other than a random forest with 100 classifiers) would be more adequate.
>   - Please note that the goal of the classification experiment was to show agreement between classifiers that use the latent representation and ones that use the conceptual representation in order to show that our explicit vectors indeed reflect the semantics of the latent vectors.  Nevertheless, we accepted the recommendation of the reviewer and performed the experiments of Section 3.2 with a simpler classifier, KNN (K=5, DIstance function is cosine).  The results, which are much better than the RF ones, are shown here.  We will add this table to the paper.
>
>     | Dataset | raw agreement| kappa coef| CES acc| LLM acc|
>     |---------|--------------|-----------|--------|--------|
>     | 20 news | 0.82         | 0.81      | 79     | 87     |
>     | ag_news | 0.92         | 0.90      | 88     | 90     |
>     | DBpedia | 0.93         | 0.92      | 94     | 94     |
>     | Ohsumed | 0.75         | 0.73      | 65     | 72     |
>     | Yahoo   | 0.75         | 0.73      | 65     | 69     |
>     | R8      | 0.92         | 0.87      | 90     | 94     |
>     | BBC     | 0.98         | 0.97      | 97     | 98     |
>
> - Besides the agreement rate of the classifiers relying on the original and the interpretable features, their accuracy is equally important, which results are delegated to the appendix.
>   - Please note that in Section 3.2 we give a detailed explanation of why the agreement results are more important.  However, we will move the accuracy results to the main paper.
> - It seems that the use of the transformed representations degrade classification performance noticeably. Some amount of performance drop is fine in exchange for more interpretable representations, but this kind of tradeoff should be made explicit and clearly articulated.
>   - We are not suggesting using the interpretable representation for performing textual tasks (such as classification) but rather using it for explanation and debugging the results of using the LLM.  To show that our explicit representation is strongly correlated with the implicit one, we show that classifiers using the two types of representation are highly correlated. The accuracy when using the two representations is also highly correlated except in one case (20 newsgroups).  However, our new KNN results show closer results.
> - the alternative baseline approaches that assign meaning to each dimension of the original latent space are too naive, i.e., the proposed CES approach has access to C*, the concept inventory tailored for the given task, whereas the alternative approach uses either a vocabulary of the most frequent words or a concept inventory that does not enjoy the benefit of being adjusted to the domain in question.
>   - The goal of DMAwords and DMAconcepts is to represent a class of methods that try to assign a meaning per latent dimension.  It would be indeed interesting to try DMAc*, which is a sort of ablation experiment where the transformation part of our method is turned off.
> - Since the automated assessment is not constrained by human labor, this kind of comparison might have been performed over more documents.
>   - Indeed we would have preferred a larger test set.  There was, however, a reason for this.  We had a requirement for a balance test set - i.e., one with equal numbers of correct LLM-based classifications and incorrect ones.  Since two of the 4 datasets had a small number of classification errors, we had to limit the test set size.
> - the applied baseline is too simple and the comparison with other existing approaches is lacking.
>   - Many of the existing approaches that create new representation attempt to transform the LLM representation to a more interpretable one and focus on showing that the new representation leads to a good or better performance.  The goal of our work is completely different.  Our goal is to convey to a human user the semantics of any given LLM representation (rather than altering it).  In order to compare to other approaches with respect to this goal, we would need to use each of the methods with a black-box LLM, generate some human-understandable semantic representation of each input sentence, test the correlation between this representation and the original latent representation, and test the comprehensibility to humans.  We believe that the work presented here is one of the first to provide such a comprehensive evaluation.  We found it extremely difficult to compare CES to alternatives using such a methodology but we hope that others working towards similar goals will build and expand the methodology introduced here.
> - We will add the first and third papers ([1],[3]) in the related work in the part on sparse transformation (3rd paragraph). Additionally, we will change “One limitation of these methods is their reliance on an embedding matrix, limiting their applicability to static models.” to “One limitation of these methods is their reliance on an embedding/dictionary matrix”.
> - The second paper Yun, Zeyu, et al. "Transformer visualization via dictionary learning: contextualized embedding as a linear superposition of transformer factors." is very interesting. It uses dictionary learning to see the transformer model as a linear superposition of transformer factors. We will cite it in the related work.

---

### Official Review · Reviewer_iTU3 · 2023-08-11

**Soundness:** 3

**Excitement:**

4: Strong: This paper deepens the understanding of some phenomenon or lowers the barriers to an existing research direction.

**Missing References:**

- a citation for DMA?

These references are not necessarily missing, but maybe they could give some further ideas since they are based on similar idea(s), i.e., "conceptualization of embeddings":

- Koc et al., 2018: Imparting Interpretability to Word Embeddings while Preserving Semantic Structure
- Sommerauer & Fokkens, 2018: Firearms and Tigers are Dangerous, Kitchen Knives and Zebras are Not: Testing whether Word Embeddings Can Tell
- Molino et al., 2019: Parallax: Visualizing and Understanding the Semantics of Embedding Spaces via Algebraic Formulae

**Paper Topic And Main Contributions:**

The authors propose to make embedding space more interpretable with respect to the model that produced them by conceptualizing the embedding space. The concept(ual) space is supposed to be more comprehensible for humans.
They motivate their work by the need for debugging embedding models (in this case LLMs) and for detecting biases, as well as for explaining decision made by systems incorporating LMs.
They propose an algorithm for conceptualization as well as a new evaluation technique to test the conceptualization.
As concept database, they use a Wikipedia graph. For labeling edges between concepts, they calculate sibling scores based on the similarity of the sibling concepts to detect "is-a" relations.
Since concepts often are hierarchical and not all down-stream tasks might need all concepts, the conceptual space can have different granularity, depending on how "deep" a concept can branch out (or how many children concepts are available) and what is needed.
The authors accomplish this by adapting the concept space iteratively to the input text.

For evaluating their method, they use both human and LLM judgements.

The authors' main contribution is a new variant of how to make embedding spaces more interpretable, together with an interesting evaluation.

**Questions For The Authors:**

General:
- Why do we need the conceptual space in the first place? Is it not "enough" or maybe even equivalent to having close (i.e. highly similar) concepts in L, which describe the text vector t in L?

Abstract:
- The first sentence in the abstract is confusing to me: One of the "main methods" for interpreting a text is to embed it? Not really, is it? It's one of the main methods to process natural language using computers etc., but for interpreting a text, humans do not map it to vectors -- probably to concepts, though!?

Section 1:
- The embeddings of small LMs like skip-gram or GloVe were also not interpretable -- did you consider these as well? (referring to ll. 36)
- ll. 49: by "understanding" the embedding space, we might get some insights on those layers, but not into the rest of the model.


2.2:
- l. 181: |C¹| = 37: if I understand correctly, this means that in the first level of the hierarchy, there are only 37 concepts in total? Or am I missing something -- it seems a rather small number for top-level concepts in Wikipedia to me.
- I am not 100% certain I understand the "selective refinement". What does "the concept with the largest weight" mean exactly?
- How is "removeP" determined? Is it simply a parameter you set at the beginning?

2.1:
- Can you please provide a URL to the "Wikipedia category directed graph"


3.:
- For the function tau, you use the string representation of the concept, correct? Might this induce some confusion for the embedding model, since there is no context for these concepts?

3.1:
- The 10 sentences you use as context, are they preceding the sentence under consideration? How are they connected to the sentence except that they are from the same article?


3.2:
- The Kappa coefficient is "relatively high" on some of the datasets, but also quite low on others, e.g., 20 news group, Ohsumed. Also, the difference in accuracy on 20 news group is pretty large, especially when compared to the other datasets. Do you have an explanation for that?
- Maybe I oversaw it -- please mention that all datasets are in English.

3.3.2:
- Please add to the table captions that the scores represent "raw agreement".

4.2:
- The plots in Figure 2 are really interesting! I just wonder what they actually mean -- the concept of "weapons" with input "government" in GPT-2 gets stronger with higher layers -- why? What was the context for "government"?

**Reasons To Accept:**

- The work is well motivated and set into context with related approaches.
- It is definitely relevant for research in embedding spaces (wrt biases, model decisions, etc.)
- I like the idea of having different granularity for the concept spaces.
- It seems that the proposed method works well and it is certainly an interesting take on embedding space interpretation, but I am not 100% certain I understood everything (see questions).

**Reasons To Reject:**

- The method needs some clarification -- see questions.
- The evaluation is interesting, but at least the results using a classifier trained on the concept space are not super convincing, since the agreement scores are not very high. When comparing humans vs. LLM ratings, the test set is very small.

**Reproducibility:**

4: Could mostly reproduce the results, but there may be some variation because of sample variance or minor variations in their interpretation of the protocol or method.

**Reviewer Confidence:**

3: Pretty sure, but there's a chance I missed something. Although I have a good feel for this area in general, I did not carefully check the paper's details, e.g., the math, experimental design, or novelty.

**Typos Grammar Style And Presentation Improvements:**

The paper is well structured and written.

Typos:

- l. 76: "that latent dimensionS correspond .." / "that A latent dimension corresponds .."


Style:

- The conceptual space C and the set of concepts C are hard to differentiate, maybe use CS for conceptual space?

---

> ### Author Rebuttal · Authors · 2023-08-28
>
> Thank you for the feedback on our work as well as the points on the related work and the typos.
>
> Following the questions in the reviews, we performed the experiments of Section 3.2 with a simpler classifier, KNN (K=5, DIstance function is cosine).  The results, which are much better than the RF ones, are shown here.  We will add this table to the paper.
>
> | Dataset | raw agreement| kappa coef| CES acc| LLM acc|
> |---------|--------------|-----------|--------|--------|
> | 20 news | 0.82         | 0.81      | 79     | 87     |
> | ag_news | 0.92         | 0.90      | 88     | 90     |
> | DBpedia | 0.93         | 0.92      | 94     | 94     |
> | Ohsumed | 0.75         | 0.73      | 65     | 72     |
> | Yahoo   | 0.75         | 0.73      | 65     | 69     |
> | R8      | 0.92         | 0.87      | 90     | 94     |
> | BBC     | 0.98         | 0.97      | 97     | 98     |
>
> About the questions you have regarding the paper:
>
> - Why do we need the conceptual space in the first place? Is it not "enough" or maybe even equivalent to having close (i.e. highly similar) concepts in L, which describe the text vector t in L?
>   - The dimensions in L were created by the training mechanism of the LLM and are not necessarily associated with human-recognizable concepts.  One can view them as a private language of the LLM.  The idea of our work is to map these implicit vectors (rather than individual dimensions) to a space that is recognizable by humans, thus helping us to understand this private language of the machine.
> - One of the "main methods" for interpreting a text is to embed it? Not really, is it? It's one of the main methods to process natural language using computers etc., but for interpreting a text, humans do not map it to vectors -- probably to concepts, though!?
>   - We agree and we will modify this sentence to: “One of the main methods for processing natural language by a computer is through embedding…”
> - The embeddings of small LMs like skip-gram or GloVe were also not interpretable -- did you consider these as well? (referring to ll. 36)
>   - Testing our method on older types of  LMs should be interesting and can be potentially added in the future.  In this work, however, we concentrated on transformer-based approaches.
> - 49: by "understanding" the embedding space, we might get some insights on those layers, but not into the rest of the model.
>   - Our main focus is not on understanding the complete model but on understanding the semantics of the input text according to the model.
> - 181: |C¹| = 37: if I understand correctly, this means that in the first level of the hierarchy, there are only 37 concepts in total? Or am I missing something -- it seems a rather small number for top-level concepts in Wikipedia to me.
>   - These are only the most abstract concepts such as Mathematics, People, Music, Ethics, Philosophy, etc. The current concepts in Wikipedia at that level can be found on this Wikipedia page https://en.wikipedia.org/wiki/Category:Main_topic_articles
> - I am not 100% certain I understand the "selective refinement". What does "the concept with the largest weight" mean exactly?
>   - To clarify this step we will add the following: “The concept with the largest weight after the projection to CES is then selected for expansion.  The intuition is that this concept represents a main topic of the text, and will therefore benefit the most with more refined representation.”
> - How is "removeP" determined? Is it simply a parameter you set at the beginning?
>   - This is indeed a system parameter.
> - Can you please provide a URL to the "Wikipedia category directed graph"
>   - The Wikipedia category-directed graph is accessible by starting with the root URL: https://en.wikipedia.org/wiki/Category:Main_topic_articles. We included the modified category graph we have extracted in the files we submitted with this submission.
> - For the function tau, you use the string representation of the concept, correct? Might this induce some confusion for the embedding model, since there is no context for these concepts?
>   - The average length of a concept name is 4.25 which is sufficient in most cases.  However, we included an additional type of \tau, called \widehat\tau (section 2.3) that adds children's concept names as a context. An experiment comparing the two can be found in Appendix D. In the future, we consider adding some text from the Wikipedia articles belonging to the concept as context.
> - The 10 sentences you use as context, are they preceding the sentence under consideration? How are they connected to the sentence except that they are from the same article?
>   - We used the first 10 sentences of the article to generate C*. The sentence we tested on was found elsewhere in the text. Since we assume that a news article is focused on a specific topic, we expect the first 10 sentences to provide an appropriate context.
> - The Kappa coefficient is "relatively high" on some of the datasets, but also quite low on others, e.g., 20 news group, Ohsumed. Also, the difference in accuracy on 20 news group is pretty large, especially when compared to the other datasets. Do you have an explanation for that?
>   - 20 Newsgroup and Ohsumed have more labels than the others, thus increasing the likelihood of disagreement. Recall that Kappa is between -1 and 1, and for example, Kappa=0.6 is considered (in general) as a good agreement.  We are not sure about the reason for the larger difference in accuracy for the 20-newgroup dataset.  However, in a new experiment included here, using KNN classifier, we observe this phenomenon less strongly.
> - Maybe I oversaw it -- please mention that all datasets are in English.
>   - We will
> - 3.3.2: Please add to the table captions that the scores represent "raw agreement"
>   -  We will
> - 4.2:The plots in Figure 2 are really interesting! I just wonder what they actually mean -- the concept of "weapons" with input "government" in GPT-2 gets stronger with higher layers -- why? What was the context for "government"?
>   - We used the word "Government" without context as if it was the start of the sentence. We believe that it could be that in the initial layers, the embedding connects to closely related concepts. But in the final layers, it extends to a broader understanding of the context and to words that should appear next to it ( it could be that GPT2 expects that next to “Government” we should see a context related to the concept WEAPON).
> - humans vs. LLM ratings, the test set is very small
>   - Indeed we would have preferred a larger test set.  There were, however, two reasons for this.  The first was limited access to human taggers.  The second is our requirement for a balanced test set - i.e., one with an equal number of correct LLM-based classifications and incorrect ones.  Since two of the 4 datasets had a small number of classification errors, we had to limit the test set size.
> - a citation for DMA?
>   - This is a term that we defined in Section 3.3.2.  It is an acronym for Dimension Meaning Assignment.
> - Koc et al., 2018: Imparting Interpretability to Word Embeddings while Preserving Semantic Structure
>   - will be added in the last paragraph of the related work as it is a work that changes the objective function of GloVe to make it more interpretable, similar to the paper “ Learning interpretable word embeddings via bidirectional alignment of dimensions with semantic concepts” that is cited in this paragraph.
> - Sommerauer & Fokkens, 2018: Firearms and Tigers are Dangerous, Kitchen Knives and Zebras are Not: Testing whether Word Embeddings Can Tell
>   - It is an interesting paper that talks about semantic features that can be abstracted from the embedding using a supervised classifier. We will add it to the related work in the second to last paragraph that discusses probing methods.
> - Molino et al., 2019: Parallax: Visualizing and Understanding the Semantics of Embedding Spaces via Algebraic Formulae
>   - It introduces a tool for doing simple operations such as PCA and t-SNE on any embedding. We will cite it in the related work.

---

### Meta-Review · Area_Chair_pCBQ · 2023-09-15

**Recommendation:** 5

**Metareview:**

The submission presents a method for mapping from embedding space into a space defined by a set of human-interpretable concepts, using textual descriptions of each concept. Evaluation is performed via text classification. The initial reviews flagged some weaknesses in evaluation, which were addressed by further evaluations during the discussion period. In the end, all reviewers rated the paper as "strong" on excitement and either "good" or "strong" on soundness.

---

### Decision · Program_Chairs · 2023-10-07

**Decision:**

Accept-Main

**Comment:**

The submission presents a method for mapping from embedding space into a space defined by a set of human-interpretable concepts, using textual descriptions of each concept. Evaluation is performed via text classification. The initial reviews flagged some weaknesses in evaluation, which were addressed by further evaluations during the discussion period. In the end, all reviewers rated the paper as "strong" on excitement and either "good" or "strong" on soundness.